# A Systematic Review and Meta-Analysis of Malaria Test Positivity Outcomes and Programme Interventions in Low Transmission Settings in Southern Africa, 2000–2021

**DOI:** 10.3390/ijerph19116776

**Published:** 2022-06-01

**Authors:** Olukunle O. Oyegoke, Olusegun P. Akoniyon, Ropo E. Ogunsakin, Michael O. Ogunlana, Matthew A. Adeleke, Rajendra Maharaj, Moses Okpeku

**Affiliations:** 1Discipline of Genetics, School of Life Sciences, University of KwaZulu-Natal, Durban 4000, South Africa; drkunleoye@gmail.com (O.O.O.); akohseg@gmail.com (O.P.A.); adelekem@ukzn.ac.za (M.A.A.); 2Biostatistics Unit, Discipline of Public Health Medicine, School of Nursing & Public Health, College of Health Sciences, University of KwaZulu-Natal, Durban 4000, South Africa; oreropo@gmail.com; 3Department of Physiotherapy, Federal Medical Centre, Abeokuta 110118, Nigeria; ogunlanam@ukzn.ac.za; 4College of Health Sciences, University of KwaZulu-Natal, Durban 4000, South Africa; 5Malaria Research Group, South African Medical Research Council, Durban 4000, South Africa; rajendra.maharaj@mrc.ac.za

**Keywords:** malaria, reactive case detection, rapid diagnostic test, PCR, Southern Africa, surveillance

## Abstract

Malaria is one of the most significant causes of mortality and morbidity globally, especially in sub-Saharan Africa (SSA) countries. It harmfully disturbs the public’s health and the economic growth of many developing countries. Despite the massive effect of malaria transmission, the overall pooled proportion of malaria positivity rate in Southern Africa is still elusive. Therefore, the objective of this systematic review and meta-analysis is to pool estimates of the incidence of the malaria positivity rate, which is the first of its kind in South African countries. A literature search is performed to identify all published articles reporting the incidence of malaria positivity in Southern Africa. Out of the 3359 articles identified, 17 studies meet the inclusion for systematic review and meta-analysis. In addition, because substantial heterogeneity is expected due to the studies being extracted from the universal population, random-effects meta-analyses are carried out to pool the incidence of the malaria positivity rate from diverse diagnostic methods. The result reveals that between-study variability is high (*τ*^2^ = 0.003; heterogeneity I^2^ = 99.91% with heterogeneity chi-square *χ*^2^ = 18,143.95, degree of freedom = 16 and a *p*-value < 0.0001) with the overall random pooled incidence of 10% (95%CI: 8–13%, I^2^ = 99.91%) in the malaria positivity rate. According to the diagnostic method called pooled incidence estimate, the rapid diagnostic test (RDT) is the leading diagnostic method (17%, 95%CI: 11–24%, I^2^ = 99.95%), followed by RDT and qPCR and RDT and loop mediated isothermal amplification (LAMP), respectively, found to be (3%, 95%CI: 2–3%, I^2^ = 0%) and (2%, 95%CI: 1–3%, I^2^ = 97.94%).Findings of the present study suggest high malaria positive incidence in the region. This implies that malaria control and elimination programmes towards malaria elimination could be negatively impacted and cause delays in actualising malaria elimination set dates. Further studies consisting of larger samples and continuous evaluation of malaria control programmes are recommended.

## 1. Introduction

Malaria is one of the most significant causes of mortality and morbidity in many developing countries [1,2,3]. It is caused by *Plasmodium* parasites [4,5]. It is projected that approximately 3.3 billion individuals have malaria worldwide [6], and malaria constitutes acritical health challenge for countries in sub-Saharan Africa (SSA) [7,8,9,10]. The World Health Organization (WHO) noted that of all the 241 million malaria cases recorded in 2020, 95% were from sub-Saharan Africa, with Nigeria accounting for 25%. The same year recorded an estimated global number of malaria deaths of 627,000, with 96% of the deaths in sub-Saharan Africa [11].

In general, the approach to malaria control has been through the use of different methods. These include, among others, the use of insecticide-treated nets (ITN), indoor residual spraying (IRS), application of chemicals to stagnant water bodies to kill immature forms of mosquitoes and the use of approved medications of which artemisinin combined therapy (ACT) is currently recommended [12,13]. These combinations have been proven to be highly effective [12,14,15,16].

As part of the measure to stem the tide of the rising malaria morbidity and mortality, the WHO introduced a new malaria programme initiative in 2012 tagged *T3,* which stands for Test, Treat and Track [17]. This initiative emphasizes *testing* with either RDT or microscopy to ascertain the accurate diagnosis of malaria; *treatment* with ACT and *tracking* emphasized putting adequate malaria surveillance measures and strengthening existing tools that could lead to significant malaria reduction and elimination [17]. Active case detection (ACD) is a surveillance technique recommended by the World Health Organisation; it gives credence to the third component mentioned, which is tracking. ACD is applicable in climes with low or deficient malaria transmission levels or where malaria elimination is focused [18]. ACD can be subdivided into Reactive Active Case Detection (RACD or RCD) and Passive Active Case Detection (PACD). With RCD, there is the identification of an index case (usually from Passive Case Detection) which initiates the active case detection among the population or households that are linked with the index case [19].

On the other hand, PACD involves early detection of possible malaria cases or transmission among target groups, especially during transmission seasons; therefore, PACD does not necessarily have to do with index case identification [18]. In some settings, RCD has been identified as a potential means to enhance malaria case detection and improve health care provision [19]. Going by this understanding and recommendations by the World Health Organization, it was evident that RCD would not be effectively applied in countries or regions with a malaria-endemic majority, which are present in sub-Saharan Africa, particularly in the west and central African regions. Although some studies have been conducted in these malaria-endemic regions on RCD [20,21,22], RCD as a form of surveillance tool has been demonstrated to be more appropriate in low endemic settings [18,23].

It is noteworthy that in Africa, significantly lower levels of malaria transmission have been demonstrated in countries in the southern Africa region, with some already working hard towards achieving elimination status [24,25,26]. In addition, RCD is widely applied in many [27,28,29], with RDT commonly used for diagnosis at the community level and the adoption of ACT [30,31]. With these, it becomes possible to calculate the test positivity rate (TPR), which is the number of laboratory-confirmed malaria tests per hundred suspected cases examined; it can be used to monitor the impact of the programmes on malaria transmission [11,32]

To date, however, the national estimate of the malaria positivity rate in the pre-elimination setting of Southern Africa based on the various diagnostic tools applied is still elusive. The systematic review and meta-analysis confirm existing evidence. The evidence may assist policymakers and programme managers in designing valid policies to control and ultimately eliminate malaria. Furthermore, no published systematic review or meta-analysis confirmed the pooled proportion of malaria positivity rate in Southern Africa. Thus, this study aims to determine the pooled proportion of the malaria positivity rate in Southern Africa from studies conducted between January 2000 and December 2021.

## 2. Materials and Methods

### 2.1. Study Area

According to the United Nations (UN) criteria, the whole continent of Africa was grouped into five sub-regions, but the current study examined one of the regions—Southern Africa. Based on the UN criteria, Southern Africa consists of eleven countries, which are Angola, Botswana, Lesotho, Malawi, Mozambique, Namibia, South Africa, Eswatini, Zambia and Zimbabwe. The island nation of Madagascar is excluded because of its distinct language and cultural heritage (Figure 1). Whereas countries such as Mozambique are known to have high incidences of malaria in the region, other countries have been keeping malaria incidence at a low transmission level and working towards elimination [24,26].

### 2.2. Search Strategy and Article Suitability

A literature search was conducted to identify all published studies reporting the prevalence or incidence of malaria in Southern Africa. The literature search strategy, selection of publications, data extraction, and the reporting of results for the review were executed following the Preferred Reporting Items for Systematic Reviews and Meta-Analyses (PRISMA) guidelines [33]. We searched PubMed, Scopus, Ebscohost, World cart, and Sabinet African Journals for relevant published articles in this systematic review and meta-analysis. These databases were searched for English articles published between January 2000 and December 2021, of which the last search was done in March 2022. During the search, no restriction was placed on the type of article, the article attributes or the study outcome. Language publications were mainly in English. The database search was done using a combination of the following terms and Boolean operators: “Malaria”, “Malaria programs”, “low transmission settings”, Malaria Test and Treat”, “Malaria Test results”, “Malaria test positivity”, “Malaria elimination tools”, “Reactive Active Case Detection”, “Reactive Case Detection”, “Approach to elimination” and “Southern Africa” (each country in Southern Africa was also individually combined)—an additional search was performed using the reference lists of these articles. The inclusion criteria include studies conducted to determine the malaria positivity, studies that made reports on the various programme interventions following the application of RCD, studies written in the English language, studies published between 2000 and 2021 and studies conducted in Southern Africa. The exclusion criteria are as follows: articles published before January 2000, articles with study design such as reviews, letters to editors, editorials, commentaries, expert opinions, books, book chapters, brief reports and theses or publications with difficulties in identifying the malaria test positivity outcome and studies conducted outside Southern Africa.

### 2.3. Data Extraction and Quality Assessment

Two authors (O.P.A. and M.O.O.) working independently were responsible for the article search and selection. This was later doublechecked by the third author (O.O.O.). The initial selection was done using the articles’ title followed by article screening using the abstract. From the resultant articles, the ones that did not further fill the above criteria were excluded. Following this, the selected articles were reviewed in detail and the information was extracted from each article. Number of incidence was determined as a relationship between total number of individuals screened and number of positives cases. The data quality assessment was done independently by two reviewers (M.O.O. and O.O.O.) using the Mixed Method Appraisal Tool (MMAT) version 2018 [34]. Because this study focused on quantitative parameters, the categories which relate to the qualitative data quality assessment were utilized: quantitative randomized controlled trials and quantitative non-randomized controlled trials. The MMAT design questions as related to the selected studies were scored accordingly as “YES”, “NO” or “CAN’T TELL”. There is a section in the tool that made provision for input of any other necessary comments. Analysis of the outcome of the quality data quality assessment is attached as Appendix A.

### 2.4. Statistical Data Analyses

Data were extracted into a Microsoft Excel spreadsheet and the final data sets were translated into system files for statistical analysis. All the statistical analyses were performed using *metaprop written* command. Statistical analyses were done using Stata17.0 (Stata Corp., College Station, TX, USA). The overall pooled incidence was computed using the DerSimonian–Laird method for the random-effects model, based on the inverse variance method for measuring weight. Inconsistency index (denoted by I^2^ statistics) was used to evaluate the magnitude of heterogeneity among studies included in the final meta-analysis, with I^2^ values > 25%, 25–75% and < 75% interpreted as low, moderate and high heterogeneity, respectively. In addition, we assessed publication bias by visual inspection of the funnel plot and further substantiated it using Egger’s regression test and Begg’s correlation test. Further, to assess the possible source of heterogeneity between studies, we did a subgroup analysis based on country, diagnostic method and publication year. Ref. [35] identified target groups, intervention types, study designs and measurement outcome as possible sources of heterogeneities in meta-analysis studies. Meta-regression was used to investigate factors potentially contributing to the between-study heterogeneity. Univariable analysis was done for each selected variable included. Population screened as a continuous variable and study year as a categorized variable were used in the final meta-regression model.

### 2.5. Results

Figure 2 shows electronic data sources and the study selection process. Following an initial combination using Boolean operators, we obtained a total of 3359 records from PubMed, Scopus, and Sabinet Africa Journal out of which 2522 were removed due to duplicates (also due to other reasons). Using the inclusion criteria, the total number of articles based on title search extracted was 837 out of which 16 qualified for abstract search. Eight of these fitted for critical reading of the full texts as per the inclusion criteria along with nine other articles that were identified from the citation search. Overall, seventeen studies were deemed eligible for inclusion by our electronic database search strategies.

### 2.6. Descriptive Results of Eligible Studies

The features of the seventeen eligible studies included in this systematic review and meta-analysis are presented in Table 1. All included studies were conducted in three diverse countries; with the highest number of included studies (*n* = 10, 52.6%) conducted in Zambia, four conducted in Eswatini (*n* = 4, 21.1%) and three conducted in Namibia (*n* = 3, 15.8%). The largest number of included studies (*n* = 7) used RDT to screen the malaria patients, whereas the remaining diagnostic method employed two or three diagnostic methods. The number screened for eligible studies ranged from 953 to 597,631. All the included studies were conducted between 2000 and 2021, with 2, 4 and 11 of the included studies conducted between 2006 and 2010, 2012 and 2015 and 2016 and 2021, respectively.

### 2.7. Meta-Analysis of the Overall Incidence of Malaria Test Positivity in Southern African Countries

Due to the anticipated variation between studies, random-effects meta-analyses were performed using the total screened and number of positives (effect size and standard error of the effect size). The meta-analysis revealed that between-study variability was high (*τ*^2^ = 0.003; heterogeneity I^2^ = 99.91% with heterogeneity chi-square *χ*^2^ = 18,143.95, degree of freedom (df) = 16 and *p*-value < 0.0001). This finding implies that the included studies share a common effect size. Individual study incidence estimates ranged from 0% to 45% with the overall random pooled incidence of malaria positivity of 10% (95%CI: 8–13%). In addition, studies weighted approximately equal with weights on individual studies ranging from 5.53% to 5.98% due to high heterogeneity between studies. Figure 3 presents the forest plot derived from the meta-analysis.

### 2.8. SubgroupMeta-Analysis

Subgroup analyses were done for countries Zambia, Eswatini and Namibia (Table 2). Subgroup analysis of pooled incidence estimate stratified by country was (16%, 95%CI: 12–21%, I^2^ = 99.93%, *p <* 0.001, *n* = 10) for Zambia, (1%, 95%CI: 1–2%, I^2^ = 94.91%, *p <* 0.001, *n* = 4) for Eswatini and (3%, 95%CI: 2–4%, I^2^ = 0.0%, *n* = 3) for Namibia. Also, subgroup analysis stratified by diagnostic method was 17% (95%CI: 11–24%, I^2^ = 99.95%, *p <* 0.001, *n* = 7), 2% (95%CI: 1–3%, I^2^ = 97.94%, *p <* 0.001, *n* = 4), and 3% (95%CI: 2–3%, I^2^ = 0%, *p <* 0.001, *n* = 2) for RDT, RDT and LAMP and RDT and qPCR, respectively (Table 3). The subgroup analysis stratified based on year 2012 was 28% (95%CI: 25–31%, I^2^ = 0.0%, *n* = 1), followed by year 2016 which was 22% (95%CI: 21–23%, I^2^ = 0.0%, *n* = 1).In addition, findings from meta-regression indicate that the overall year of publication was significant, but the result from subgroup analysis reveals that only studies conducted in year 2017 is not significant using the 95% confidence interval. Results of the subgroup analysis are depicted in Figure 4, Figure 5 and Appendix A, respectively. Subgroup analysis was also performed for year of publication (Figure 6).

### 2.9. PublicationBias Assessment

Furthermore, we evaluated publication bias (small-study effects) on our pooled incidence estimates through visual inspection of the funnel plot. Based on the outcome of the qualitative evaluation of funnel plot symmetry, it seems there is no evidence of publication bias (Figure 6). Therefore, to quantitatively establish the findings of the funnel plot, we also performed Egger’s regression test and Begg’s correlation test. Both Egger’s regression test (*p* = 0.365) and Begg’s correlation test (*p* = 0.973) showed the absence of publication bias because the *p*-value is not statistically significant. In addition, we performed the trim and fill method and there were no filled studies that pinpointed the lack of detectable publication bias.

### 2.10. Meta-Regression Analysis

Meta-regression analysis was done for each variable included in the study separately. The variables included were population screened as a continuous variable and study year as a continuous variable. Those variables with *p*-values less than 0.4 were used in the multivariable meta-regression analysis. Continuous variable population screened variable and study year variable had significant value and retained it in the final multivariable analysis. Results of the final multivariable meta-regression are summarized in Table 4 and Figure 7. As revealed in Table 4, the between-study variability was high. This variation may be due to different studies linking a different diagnostic method and different populations screened. Thus, the findings of meta-regression affirmed that effect size estimates were significantly predicted by the study year reported. This is an indication of expecting higher variation between studies.

## 3. Discussion

Despite the decline of malaria globally, the disease is still a major public health concern, and it is one of the leading causes of morbidity and mortality in Southern Africa. To the best of the authors’ knowledge, this is the first meta-analysis of pooled malaria test positivity outcomes in Southern Africa, reporting literature from 2000 to 2021 obtained through analysis of a systematic review and meta-analysis that pooled 3359 published articles on malaria. Therefore, it is vital to communicate the status of pooled malaria test positivity outcomes to estimate the total effect and set out control strategies for successfully eliminating the menace. However, the number of studies to be included in the final meta-analysis has been significantly reduced due to heterogeneous literature, inappropriate study designs, unrepresentative sample size and lack of data on the required indicators and other inter-related indicators.

This study affirms that the positivity is low in the low transmission settings of Southern Africa. Previous systematic reviews and meta-analyses investigating the malaria test positivity outcomes generally, and particularly in Southern Africa, are lacking. However, from individual studies, [32] reported a test positivity rate of 4.5–59% in a rural setting of Uganda, an area with high transmission whereas [52]) noted a steady test positivity reduction in the Mandla District of India, which currently has a very low transmission level. Irrespective of the setting, it has been demonstrated that the test positivity rate is an appropriate proxy of transmission intensity [53].

The outcome of these seventeen studies showed that the overall pooled positivity of malaria was 10% (95%CI: 8–13%). A possible reason for this relatively high value from these low transmission settings could be that most of the studies analysed were from Zambia, which accounted for 16% (95%CI: 12–21%) according to the subgroup meta-analysis. Although the southern part of Zambia is known for low malaria transmission, malaria is still endemic in the northern and eastern parts of Zambia [54]. For Eswatini and Namibia, the subgroup pooled positivity rates were (1%, 95%CI: 1–2%) and (3%, 95%CI: 2–4%), respectively, and these are in keeping with the WHO expected outcome in pre-elimination settings (positivity < 5% in peak malaria season) [55]. In the same vein, the [54] indicated the malaria incidence rate to be 0.8 per 1.7 per 1000 population for Eswatini and South Africa, respectively. These are countries recognised to be in the pre-elimination stage in the Southern Africa region.

In general, the main interventional approach applied in most malaria-endemic settings includes the use of insecticide-treated nets (ITNs), indoor residual spray (IRS), intermittent preventive treatment of malaria in pregnant women and case management based on approved standard guidelines [56,57]. However, other documented programmatic interventions include larva control and vector control mass drug administration (MDA) [58,59].

According to [2,60], ITN and IRS have played key roles in reduction of malaria levels in SSA, which has led to a 40% fall in the incidence of clinical diseases. Furthermore, the focus of this current study is mainly on interventions in malaria low-endemic areas where the use of ITNs and IRS have been well promoted and applied over the years, resulting in low transmission as also noted in the analysed reports [41,48,50,51].

MDA, focal MDA (fMDA) and reactive focal MDA (rfMDA)in the background of detection methods such as RACD were the methods applied in four of the studies analysed [41,48,50,51]. The choice of these interventional approaches was based on the low transmission level of malaria in the settings where the studies were conducted—southern Zambia, Eswatini and Namibia.

Case management evaluation was the main focus of some of the other studies that were included in this analysis. Before the widespread use of ACTs, [36] looked at the efficacy of atovaquone–proguanil (AP) and sulphadoxine–pyrimethamine (SP) as interventional drugs in malaria anaemic children using the randomised, double-blind, controlled trial. AP was administered to 128 children with a packed cell volume of 9% and P. falciparum parasitaemia whereas 127 children had treatment with SP. A significant amount of treatment failure (22%) was recorded among the group who received SP and 10% among those who had AP (OR: 3.34, 95% CI: 1.54, 7.21). Ten children had a blood transfusion and six deaths that were not medication-related (five from the AP group) were noted. The study concluded that intervention by AP was more effective than SP.

Ref. [38] assessed the possibility of enhanced community delivery of AL through increased capacity building of community health workers (CHW) in Southern Zambia. This was borne out of the simplicity of RDT use as well as the fact that with minimal on-the-job training CHWs can be an easy avenue to transfer malaria interventional programmes to the wider community [61,62]. Overall, the study reported that there was a high level of treatment accuracy (94–100%) based on training received by the CHWs [38]. In another study by [37], which assessed the therapeutic efficacy of AL to distinguish parasite recrudescence from re-infections, it was noted that there was significant gametocyte reduction among 80% of the participants that were successfully followed up with in the study. Follow up after 28 days showed an adequate clinical and parasitological response of 100%, with late parasitological failure being a result of re-infection rather than recrudescence [37].

The effectiveness of dihydroartemisinin–piperaquine (DHAp) in MDA based malaria intervention programmes were tested in three of the studies [41,50,51] whereas artemether-lumefantrine and single-dose primaquine were used in the study by [48]. TheDHAp regimen was proven to have a high level of acceptance and adherence (MDA—80.6–84.9%; fMDA—91.5–93.3%; rfMDA 98.7%) with no serious adverse effects [41,50,51] and good parasite clearance—100% (day 3), 97.4% (day 7), 97.8% (day 30) and 95.8% (day 60) [50]. According to [41], a significant reduction of 87% in malaria prevalence was noted following MDA in the study areas with the low transmission and a 75% reduction in malaria incidence when rfMDA was combined with RAVC [48], whereas [51] reported 48% local malaria incidence with the use of rfMDA compared with RACD. RRACD as a malaria surveillance strategy recommended by WHO is a tool that seeks to detect or identify the malaria parasites among individuals who live with or close to people who are identified as index cases [18,49]. It is a surveillance tool that has found application in malaria hotspot areas. This is based on the fact that as malaria level decreases in a community, the asymptomatic carrier proportion of the infection increases and the location, as well as the localisation of infection transmission, become spatially characterised. This indirectly creates an opportunity for healthcare accessibility [19,43,44,45,63].

RACD is practised in both malaria low transmission level settings and malaria-endemic regions; however, it is a tool that tends to be more appropriate in low transmission settings with the outcome depending on different associated factors. Most empirical studies that made use of RCD do not have a consensus radius at which malaria parasite positivity becomes most effective [64] and this is an observation that was equally noted in our study. Test positivity outcome is dependent on various other factors aside from the screening radii [44].

RDT was the most common diagnostic tool applied during most documented RACD operations and it was the main diagnostic tool used by the articles included in this study [19,39,42,43,44,45,46,47,48,49]. It is one of the recommended malaria diagnostic tools by the World Health Organisation [17,65,66,67]. It has the advantage of quick turn-around time and is easily applicable in most settings [66,68]. Other tools which have been applied include microscopy, LAMP and PCR; each of these has its own merits and demerits [43,45,46,47,49].

There are individual studies in which the sensitivity of the diagnostics used was discussed and RDT was used as a baseline diagnostic testing tool [43,44,47,63,69]. In each reported instance, the sensitivity of RDT was measured against LAMP and/or qPCR. The low or limited sensitivity yield by RCD in terms of the level of parasite positivity was commonly reported too [43,45,47,48,63,69]. RDT sensitivity has been considered a major limitation to the effectiveness of RACD as a surveillance tool by some [46,69]. The reported RDT sensitivity yield from selected articles ranged between 9.3% and 40% [45,47]. In instances where the sensitivity of LAMP was measured against qPCR, LAMP exhibited a high level of sensitivity—95% [45]. When compared with RDT, the findings showed that LAMP and qPCR detected more parasite cases among screened clusters that are asymptomatic [43,45,46,48,63,69]. Despite the low sensitivity, it has been demonstrated that RDT used in RACD could prevent 29–50% of the infection transmission from human to mosquito [43], although the contrary was noted in another study, which observed that RCD is not likely to prevent the transmission [69].

The findings from our grouping analysis based on types of malaria diagnostic tools—RDT, LAMP, and qPCR—showed that the proportion of malaria positivity outcomes among the community screened was unequally distributed across the three tools examined in the current meta-analysis. These variations could be explained by different reasons which include the fact that some of the studies were conducted in different malaria transmission seasons; some were conducted during the high transmission periods whereas others were conducted during the low transmission seasons [70], which could be a significant contributing factor for the high variations [71]. Nevertheless, this review has contributed empirical evidence with regards to the proportion of malaria positivity tests among the population screened, and some limitations can be addressed in future systematic reviews and meta-analyses.

Meta-regression is used in systematic quantitative analysis to pool covariate data [72]. It calculates the variation extent from different factors such as target populations and groups, intervention types, study designs and measurement outcomes that could introduce heterogeneities [35]. This meta-regression made use of year of publication and population screened separately. Population screened and study year variables had significant values (*p* < 0.5) and between-study variability was high, which can be attributed to different studies linking different diagnostic methods and different populations screened. Thus, the finding of meta-regression affirmed that effect size estimates were significantly predicted by the study year reported.

## 4. Limitation

This study summarized the proportion of malaria positivity tests in low transmission settings of Southern Africa, which are thus relevant in policymaking; however, it has some limitations. First, the small number of articles involved in this systematic review and meta-analysis could affect the overall proportion estimate. Second, almost 60% of the involved articles were obtained from Zambia, whereas the others were found in Eswatini and Namibia with some countries in the region having no representative data. This unequal distribution of articles throughout the region may affect the outcomes of this study. These limitations mentioned above might affect the findings conveyed in this systematic review regarding the pooled proportion of malaria positivity rate among the population screened based on the articles used.

## 5. Conclusions

The current study used systematic review and meta-analysis to gather and analyse the evidence on malaria positivity and used meta-analysis to determine the malaria positivity rate in Southern Africa. It stands as a source of valuable information for policymakers and administrators in the Southern Africa region. MDA, fMDA and rfMDA were the main programmatic methods applied with RDT being the most common diagnostic in spite of its limited sensitivity. The outcome of positivity using grouping analysis of the different diagnostic tools showed variable outcomes. Among the interventional medication regimen, DHAp was the most widely accepted, showing a high level of adherence among the participating populace. The findings in this present study suggest that the pooled estimate proportion of malaria positivity tests among the community screened was relatively high and not in keeping with the recommendation of the WHO; although, the outcome of individual subgroup meta-analysis well represents the low transmission of malaria found in the analysed countries, with the applied diagnostic tools showing a varying positivity rate. The findings from this study imply that malaria intervention programmes have had success in the reduction in malaria positivity and incidences in some regions; however, high positivity in the majority of the regions could underplay malaria elimination efforts. Furthermore, imported malaria cases across unrestricted borders from high to low transmission zones would result in longer period for malaria elimination efforts to be realized. Future focus on studies involving larger sample size, continuous and consistent evaluation of programme impacts on malaria positivity and reduction across malaria endemic regions is suggested. Improving upon the already-achieved malaria elimination gains requires the sustenance of the currently effective system with policies that give strong consideration to a wider coverage of malaria programmes using more sensitive and easy-to-use diagnostic methods.

## Figures and Tables

**Figure 1 ijerph-19-06776-f001:**
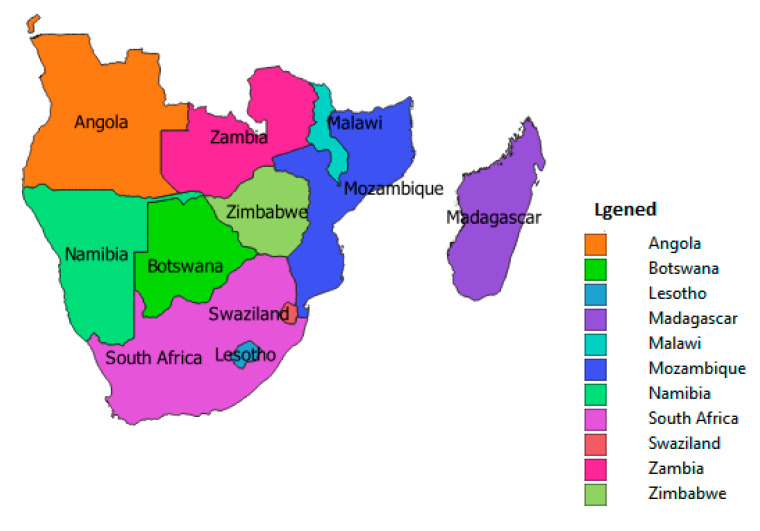
Geography of Southern Africa.

**Figure 2 ijerph-19-06776-f002:**
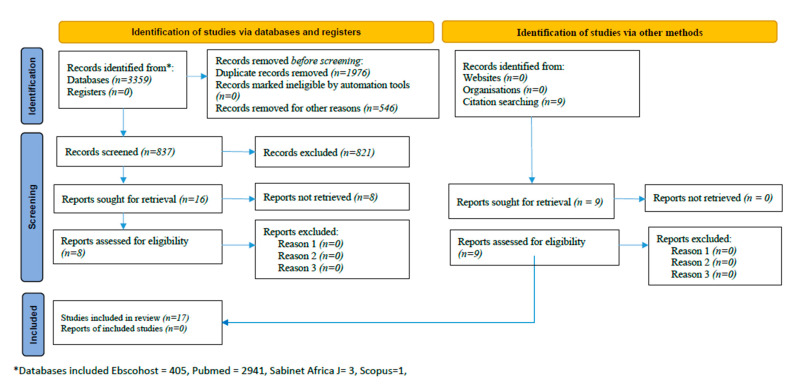
Flow diagram of literature search showing study selection process implemented for this meta-analysis.

**Figure 3 ijerph-19-06776-f003:**
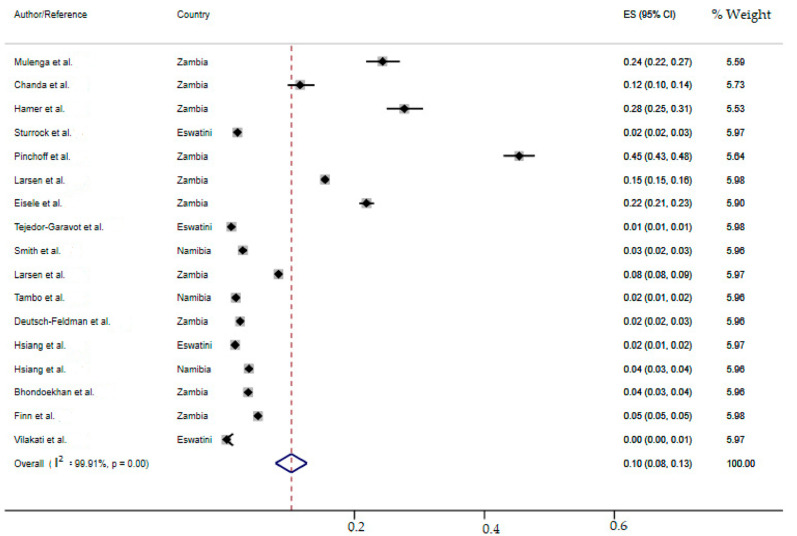
Forest plot of malaria test positivity incidence estimates in Southern Africa [19,36,37,38,39,40,41,42,43,44,45,46,47,48,49,50,51].

**Figure 4 ijerph-19-06776-f004:**
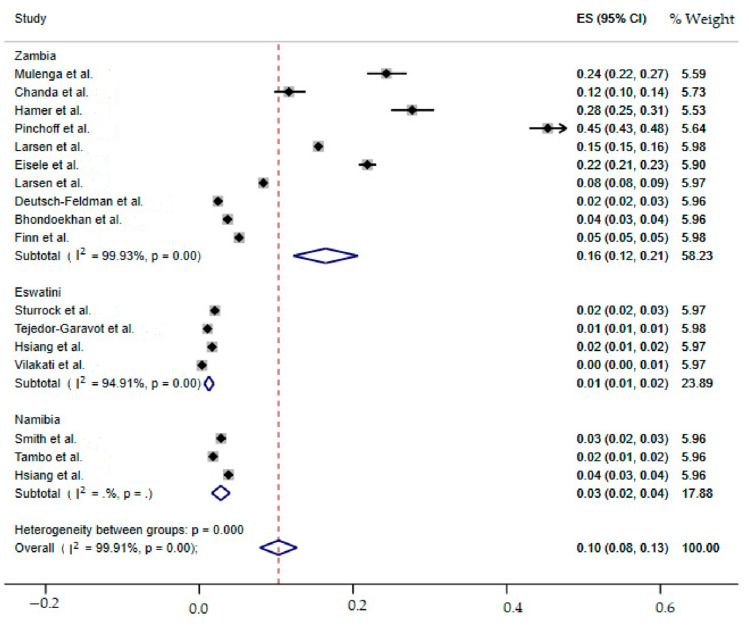
Forest plot of subgroup analysis by country [19,39,40,41,42,43,44,45,46,47,48,49,50,51].

**Figure 5 ijerph-19-06776-f005:**
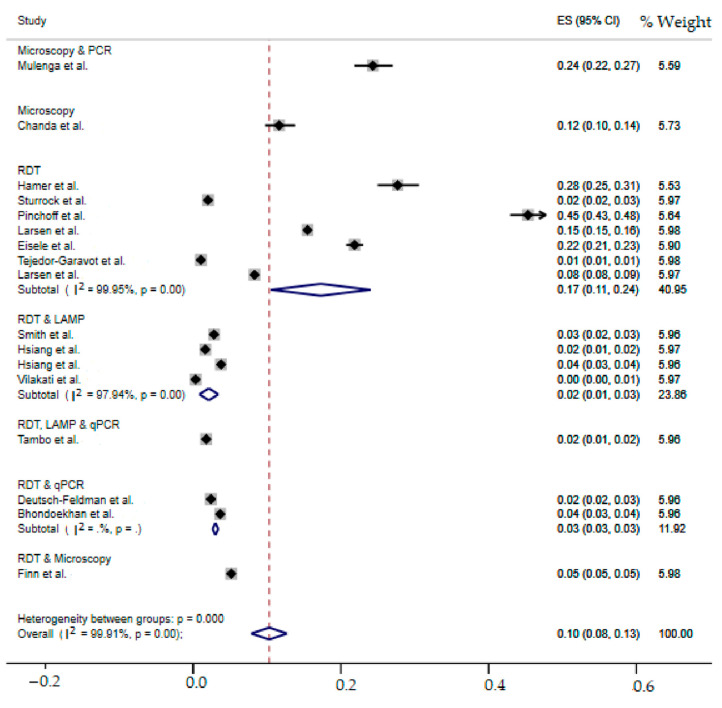
Forest plot of subgroup analysis by the diagnostic method [19,47,48,49,50,51].

**Figure 6 ijerph-19-06776-f006:**
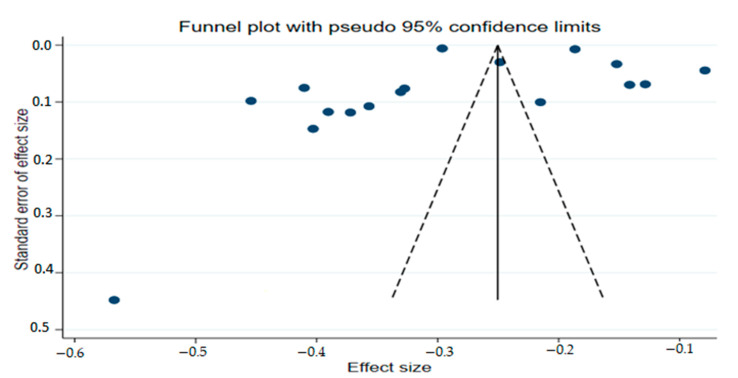
Funnel plot of the arcsine transformed incidence estimates of malaria positivity test in Southern Africa.

**Figure 7 ijerph-19-06776-f007:**
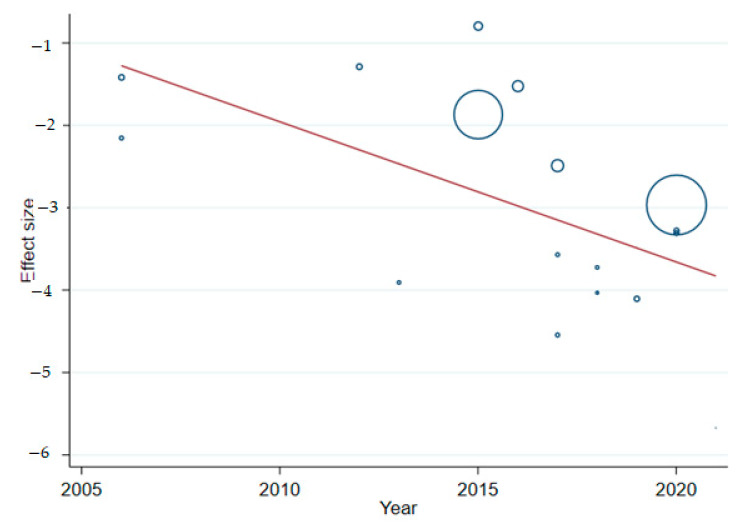
Meta-regression analysis showing trends of malaria.

**Table 1 ijerph-19-06776-t001:** Characteristics of included studies.

Author	Year	Country	Diagnostic Method	Study Type	Screened	Malaria Positive
Mulenga et al. [36]	2006	Zambia	Microscopy and PCR	Double-blind randomized control trial	1048	255
Chanda et al. [37]	2006	Zambia	Microscopy	Prospective study	953	111
Hamer et al. [38]	2012	Zambia	RDT	Cluster randomized control trial	975	270
Sturrock et al. [39]	2013	Eswatini	RDT	Cohort study	3671	74
Pinchoff et al. [40]	2015	Zambia	RDT	Cohort study	1621	735
Larsen et al. [19]	2015	Zambia	RDT	Descriptive cross-sectional study	143,295	22,201
Eisele et al. [41]	2016	Zambia	RDT	Cluster randomized control trial	5018	1097
Tejedor-Garavot et al. [42]	2017	Eswatini	RDT	Ecological study	9859	105
Smith et al. [43]	2017	Namibia	RDT and LAMP	Prospective case-control study	3151	89
Larsen et al. [44]	2017	Zambia	RDT	Retrospective cohort study	14,409	1200
Tambo et al. [45]	2018	Namibia	RDT, LAMP and qPCR	Prospective case-control study	2642	47
Deutsch-Feldman et al. [46]	2018	Zambia	RDT and qPCR	Prospective observational study	3016	73
Hsiang et al. [47]	2019	Eswatini	RDT and LAMP	Prospective observational study	10,890	180
Hsiang et al. [48]	2020	Namibia	RDT and LAMP	Cluster randomized control trial	4701	178
Bhondoekhan et al. [49]	2020	Zambia	RDT and qPCR	Cross sectional study	4170	153
Finn et al. [50]	2020	Zambia	RDT and Microscopy	Cluster randomized control trial	597,631	30,898
Vilakati et al. [51]	2021	Eswatini	RDT and LAMP	Cluster randomized control trial	1455	5

**Table 2 ijerph-19-06776-t002:** Subgroup analysis for comparison of malaria test positivity in different countries that met the final meta-analysis.

Country	Incidence (95%CI)	I^2^(%)	Heterogeneity Statistic	Heterogeneity Test	
				df	*p*-Value
Zambia	16(12–21)	99.93	13,227.55	9	0.00
Eswatini	1(1–2)	94.91	58.92	3	0.00
Namibia	3(2–4)	-	-	2	-
Overall	10(8–13)	99.91	18,143.95	16	0.00

**Table 3 ijerph-19-06776-t003:** Subgroup analysis for comparison of malaria test positivity of different combinations of diagnostic methods that met the final meta-analysis.

Diagnostic Method	Incidence (95%CI)	I^2^(%)	Heterogeneity Statistic	Heterogeneity Test	
				df	*p*-Value
Microscopy and PCR	24(22–27)	-	-	0	-
Microscopy	12(10–14)	-	-	0	-
RDT	17(11–24)	99.95	12,896.94	6	0.00
RDT and LAMP	2(1–3)	97.94	145.72	3	0.00
RDT, LAMP and qPCR	2(1–2)	-	-	0	-
RDT and qPCR	3(2–3)	-	-	1	0.00
RDT and microscopy	5(5–5)	-	-	0	-
Overall	10(8–13)	99.91	18,143.95	16	0.00

**Table 4 ijerph-19-06776-t004:** Final meta-regression model to assess sources of heterogeneity.

Variables	Coefficient	*p*-Value	95% Confidence Interval
Year of publication	−0.184	0.013	(−0.322, −0.045)
Population screened	1.70 × 10^−6^	0.403	(−2.51 × 10 ^−6^, 5.90 × 10^−6^)
Constant	367.042	0.013	(88.482, 645.602)

## Data Availability

Accessibility of data are in the original manuscript and the Appendix A.

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
