# Peer review of "A Systematic Review and Meta-Analysis of Malaria Test Positivity Outcomes and Programme Interventions in Low Transmission Settings in Southern Africa, 2000–2021"

_ijerph, 2022, doi:10.3390/ijerph19116776_

Round 1

Reviewer 1 Report

Dear authors,
Please find comments as in the attached file. Thank you and best wishes.

Author Response

REVIEWER 1

  1. From the introduction part, WHO initiative is on Test/Treat/Track which I think is the authors intention of putting “program intervention” in the title. However, the aim of SR and MA was only on the “test” part and it was reflected in the result section. Please find a more appropriate term/s to reflect the ‘test‘part in the title

The authors appreciate the suggestion given however the authors believe that the title is a reflection of the content of the review, and would appreciate if the title is retained as it is.

  1. Revise the conclusion to consist of 1) study finding 2) implication of study finding 3) possible future approaches

Thank you for the suggestion. Study findings, implications and future approach entered in the conclusion has been summarised and included in the abstract (line 35-39)

  1. Suggest to omit this and revise to other exclusion criteria as this is a mirror image of inclusion criteria

The suggestion to omit the year in this section is appreciated, however the authors believe that if retained it will give clarity on period for exclusion criteria determination more clearly to the readers especially to lay-readers.

  1. Can the authors clarify and explain how was incidence rates defined/selected particularly for a meta-analysis? Especially for study designs which reported each incidence rate for
  • Treatment/control arm (for RCT) and
  • Incidence rate for case/control (case control study design)

The authors appreciate this observation by the reviewer. Number of Incidence was determined as a relationship between total screened and number of positives cases. This has been included under data extraction and quality assessment (line 145-146)

  1. Please comment on the possibility of study designs that can cause heterogeneity. Strongly suggest for authors to check on this with subgroup analysis ----line 162

The authors appreciate the reviewer for making a note of comment. The observation has been addressed in the manuscript under statistical data analysis session (line 166-168)

  1. Please detail out, ?double blinded RCT -----Mulenga table 1

Thank you for this detail observation. The authors have detailed out the study types in the table 1 (line 194-195)

  1. Please detail out ??prospective cohort? Revise and recheck all study designs for clarity.

Thank you for this detail observation. The authors have detailed out the study types in the table 1 (line 194-195)

  1. The authors need to explain in method section, how was the incidence rate chosen and included. Especially for study design such as RCT and case control where incidence rate for each treatment/control (case /control) arm was reported

Thank you for this observation, this has been included under data extraction and quality assessment (line 145-146)

  1. Discussion ---line 251

Thank you for this observation, the typo-error has been noted and correction made accordingly in line 256

  1. Discussion for meta-regression findings ---line 389

The authors agree that the inclusion of discussion on the meta-regression is necessary to be added and appreciate the reviewer for this observation. A paragraph has been added commenting on the meta-regression (Line 387-395)

  1. The conclusion part needs to focus on the study findings and future approach. Please rearrange and adjust accordingly. Statement such as limited number of studies and small sample sizes should be in study limitation

The authors sincerely appreciate the reviewer for this observation. Findings, implication of findings and future perspective has been included in the conclusion section (line 410-431)

Reviewer 2 Report

Authors made a substantial revision. Authors found that the year of the sutdy confound the pooled prevalence; however, author was not performed the subgroup analysis by year further.

My recommendation is authors should perform the subgroup analysis by year and discuss accordingly.

Good luck.

Author Response

REVIEWER 2

Authors made a substantial revision. Authors found that the year of the sutdy confound the pooled prevalence; however, author was not performed the subgroup analysis by year further.

My recommendation is authors should perform the subgroup analysis by year and discuss accordingly

The authors appreciate this important suggestion which we trust would improve the quality of the manuscript. Subgroup analysis by year is now included as supplementary figure S1 in supplementary materials and comments and discussion entered in main article in lines 218-221 and 285-286.

This manuscript is a resubmission of an earlier submission. The following is a list of the peer review reports and author responses from that submission.

Round 1

Reviewer 1 Report

Reviewer Comments for Author

This interesting study that presumably is the proof of concept A Systematic Review and Meta-Analysis of Malaria Test Positivity Outcomes and Programme Interventions in Southern Africa, 2000–2021”. In this review author have summarized a systematic and meta-analysis of malaria test positivity outcomes and programme interventions in Southern Africa. There are couple of minor issues that have been mentioned below. Overall, it’s an important and well-planned study. I am recommending to accept this manuscript for publication in this journal after revision of the following issues.

Some issues:

  1. Material and Method section 2.1 authors can elaborate with more details.
  2. Material and Method section 2.3 authors can elaborate data extraction and quality assessment with more details.
  3. Results section 3.2 authors can elaborate with more details.
  4. Carefully check typo-errors in reference.

Reviewer 2 Report

Dear authors
Thank you for the opportunity to review your work. Kindly find the attached documents for comments.

Generally the title does not reflect the content of the review especially the "programme intervention" as it was not addressed at all.

Abstract: as commented

Main text: I read the introduction section with interest. However, the result section is questionable especially, i) included articles with study designs that cannot measure prevalence or incidence rates and ii) inconsistent values in Table 1, Figure 3, 5, 6, 7

Therefore, the overall findings, discussion and conclusion are affected.

Reviewer 3 Report

I have the following comments;

  1. (PRISMA) guidelines are updated in 2020. Authors should follow the updated guidelines for reporting systematic reviews.
  2. The major concern about the manuscript is the search terms used in the study were too narrow such as “Malaria elimination tool”, “Reactive Active Case Detection”, “Reactive Case Detection”, “Approach to elimination”, “Southern Africa” which some studies might be missed that why the results of the search showed only 48 records. My recommendation is author should use more broad terms by consulting with experts in malaria intervention for specific keywords. Then, the systematic review could be further performed.
  3. There must be an analysis for identifying the source (s) of heterogeneity of the pooled proportion such as subgroup analysis of the country and etc.

Author Response to reviewer 1, 2 and 3
author response. pdf